# Active Control of Regenerative Brake for Electric Vehicles

**Chun-Liang Lin \*, Hao-Che Hung and Jia-Cheng Li**

Department of Electrical Engineering, National Chung Hsing University, Taichung 402, Taiwan;
or091234@hotmail.com (H.-C.H.); s26813580@gmail.com (J.-C.L.)
**\*** Correspondence: chunlin@dragon.nchu.edu.tw; Tel.: +886-4-2285-1549

**Abstract:** Looking at new trends in global policies, electric vehicles (EVs) are expected to increasingly replace gasoline vehicles in the near future. For current electric vehicles, the motor current driving system and the braking control system are two independent issues with separate design. If a self-induced back-EMF voltage from the motor is a short circuit, then short-circuiting the motor will result in braking. The higher the speed of the motor, the stronger the braking effect. However, the effect is deficient quickly once the motor speed drops quickly. Traditional kinetic brake (i.e., in the short circuit is replaced by a resistor) and dynamic brake (the short circuit brake is replaced by a capacitor) rely on the back EMF alone to generate braking toque. The braking torque generated is usually not enough to effectively stop a rotating motor in a short period of time. In this research task, an integrated driving and braking control system is considered for EVs with an active regenerative braking control system where back electromagnetic field (EMF), controlled by the pulse-width modulation (PWM) technique, is used to charge a pump capacitor. The capacitor is used as an extra energy source cascaded with the battery as a charge pump. This is used to boost braking torque to stop the rotating motor in an efficient way while braking. Experiments are conducted to verify the proposed design. Compared to the traditional kinetic brake and dynamic brake, the proposed active regenerative control system shows better braking performance in terms of stopping time and stopping distance.

**Keywords:** motor; braking control; regenerative brake; driver design; pulse-width modulation

## 1. Introduction

Owing to the technological advances, the Earth's environment becomes more and more serious. To reduce gasoline-vehicle-induced air pollution, it has been a trend for many countries to advocate the development of electric vehicles (EVs) with a strong endeavor. In the past, the development of electric vehicles was limited by motor driving technology and insufficient battery power so that such vehicles could not compete against gasoline ones. In recent years, the significant progress of the development of electric vehicles has been made, but most engineers focused only on how to increase energy efficiency or to improve the performance of motor propulsion. Comparatively few research tasks have been dedicated to improve braking operations by the driving motors. Drum brake, disc brake, and antilock braking system (ABS) are commonly adopted in the conventional gasoline-powered vehicles or electric vehicles. During the downhill road sections, the genre of gasoline-powered vehicles has inherent engine braking, which generates retarding forces in the engine to slow down a running vehicle, but this is definitely not for electric vehicles. While lacking such advantage, electric vehicles could still use magnetic field produced by the field coils of the motor, i.e., the back EMF to generate an analogous effect [1,2].

Most studies revealed in the published literature focus mostly on renewable technology for electric vehicles. In References [3–7], the researchers focused on the development of the wide variety of designs for electric braking control. Among these studies, the paper in Reference [7] presented an electromagnetic brake design with the regenerative braking, which directs the back EMF of the motor to the battery for energy recharge. The way can be used to prolong the driving distance of EVs. The paper in Reference [8] studies the braking sense and its consistency in the electro-hydraulic composite braking system. Aimed at this, the researchers mainly design the braking sense consistency controller that can make up the difference of braking force while keeping the braking force change rate unchanged. In Reference [9], this study proposes a new braking pressure coordinated control system with an electro-hydraulic braking function. It realizes efficient energy recovery and ensures braking safety, while considering the disadvantages of control complexity and functional limitations of existing electro-hydraulic system. In Reference [10], an electric-hydraulic hybrid drivetrain incorporating a set of hydraulic systems is proposed for application in a pure electric vehicle. This system aims at absorbing high power braking energy as well as suppressing the impact of current on the battery. The paper in Reference [11] has suggested a model-free Q-learning approach to nonlinear state-feedback anti-lock braking system (ABS) slip control. The design is a model-free tire slip control for a fast and highly nonlinear ABS. In Reference [12], the researchers design an adaptive controller to solve nonlinear dynamics and parametric uncertainty of the control of an ABS.

While many research efforts have paid attention to tackle a similar problem, most researchers focused on the use of regenerative energy to recharge battery while obtaining limited effect of wheel brake. From practical on-road experiments, we have found that the effect is not significant or even vanishes quickly once the motor speed drops. Traditional kinetic brake (i.e., short-circuiting the motor is replaced by a resistor in the current conduction loop) and dynamic brake (as kinetic brake but the resistor replaces by a capacitor) relying on the back EMF alone is usually not enough to completely halt a rotating motor in the short period of time. Our research task proposes an improved method to tackle the problem.

The paper in Reference [13] proposed an identification technique of the parameters that are needed to minimize power consumption of a city-bus equipped with permanent magnet synchronous motors (PMSMs). The MSMs have no torque ripple at commutation, possess higher torque (comparing to brushless DC motor (BLDCM)), and have the potential for extensive use in the future [14,15]. On the basis of this advantage, we select PMSMs as the test object in this research. The paper in Reference [16] proposed a regenerative braking system that considers the effects of single switch and double switch that can recycle a relatively large amount of energy to extend the life cycle of electric vehicles. In References [1,2], short circuit brake has been explored; the studies proposed that the under-bridge short circuit braking causes a complete consumption of the back EMF in the motor coils and enables dynamic braking simultaneously. Our proposed design here is different from the conventional approaches that apply only the back EMF to generate braking force. Our proposed system resolves the problem by introducing an energy boost mechanism to reduce the performance dropping rate in the conventional regenerative braking design. In addition, the motor driver and brake driver share the same inverter that meant low cost. Theoretical analysis is developed in this paper and real-world experiments are conducted to verify the proposed design, in comparison to the traditional approaches.

## 2. Active Braking Control

The proposed braking control system considers back EMF regenerated by a free running motor and uses it to generate braking torque in an efficient way. The design stores the back EMF in a pump capacitor, which is connected to the battery in a series-parallel configuration to instantaneously increase braking force and extend the acting time duration. The back EMF controlled by pulse-width modulation (PWM) technology is fed back to the capacitor as an extra energy source cascaded with the vehicle battery to provide excessive braking torque to stop the rotating motor.

To explain, consider Figure 1, which reveals the Ampère's circuital law for a DC motor during driving or braking with *F* denoting the force generated by the motor. When the electric vehicle is operating in the driving mode, force direction of the rotor's magnetic field is clockwise. Force direction of the rotor's magnetic field becomes counter-clockwise when the electric vehicle works in the braking mode.

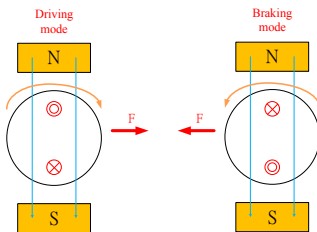

**Figure 1.** Ampère's circuital law for motor driving and braking.

The proposed system includes an active regenerative braking controller in the conventional architecture, as shown in Figure 2. It involves two metal-oxide-semiconductor field-effect Transistor (MOSFET) switches ($S_7$ and $S_8$), which are switched to charge the pump capacitor and connected the capacitor to the battery in a series-parallel configuration. The capacitor serves as a charge pump that stores the back EMF for energetic charge storage and the braking resistor consumes energy when the capacitor is being charged.

Activation of the system is controlled by braking commands. When the circuit is in operation, PWM1 ($S_1$), PWM3 ($S_3$), and PWM5 ($S_5$) only activate as a buffer. They are disabled to lead the braking current that goes through the body diode of MOSFET. Referring to Figure 3, the outputs of $S_1$, $S_2$ and $S_3$ are combined and transmitted to the photocoupler (TLP) to control $S_8$. Furthermore, the signal is sent to an inverter. The inverter reverses the signal, if the input is one, the output becomes zero. The output of the TLP is used to control $S_7$.

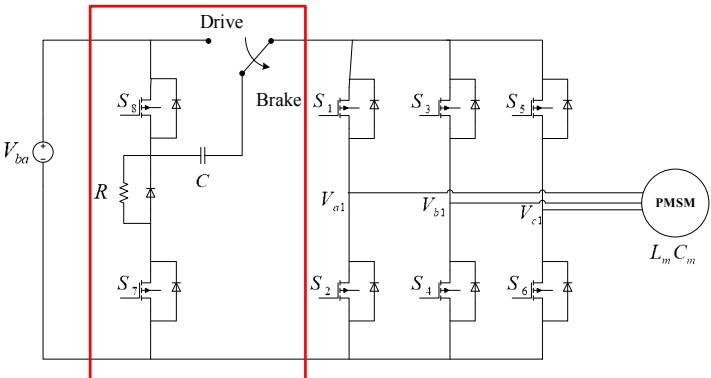

**Figure 2.** Active regenerative braking control scheme.

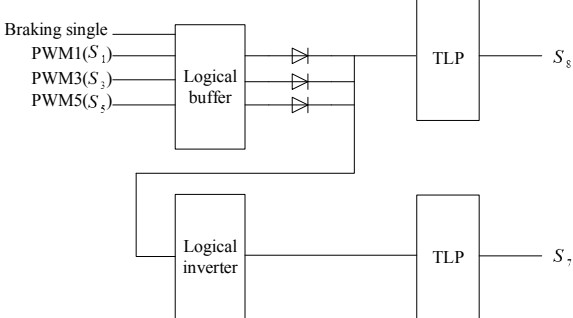

**Figure 3.** Control of MOSFET switches $S_7$ and $S_8$.

For the braking mode, the relay that appeared in Figure 2 is inactivated and the driving current loop is cut off. The microcontroller unit (MCU) turns on the MOSFET $S_7$ and turns off $S_8$ to let the back EMF generate the current $i$. Referring to Figure 2, the back EMF is rectified to the dc voltage via the body diode of the MOSFET $S_7$ where, in the figure, the pump capacitor's voltage is $V_c$, the battery voltage is $V_{ba}$, the voltage across $S_7$ and $S_8$ are, respectively, $V_{s7}$ and $V_{s8}$, the braking resistor is $R$, the motor internal resistance is $R_m$, the motor internal inductance is $L_m$, and the motor internal capacitance is $C_m$.

The controller shown in Figure 3 controls switching status of $S_7$ and $S_8$. The controller is activated and deactivated by the braking signal. When the controller works PWM1 ($S_1$), PWM3 ($S_3$), and PWM5 ($S_5$) are combined to control a logical buffer. The outputs of the logical buffer are combined and used to isolate the breaking signal from other signals. It is transmitted to a TLP to control $S_8$. Furthermore, the signal is used to activate a logical inverter and then connected to a TLP to control $S_7$.

To create an effective way to brake the PMSM in the shortest time, we first adopt the Hall table of the switching sequence of six MOSFETs for controlling the reverse magnetic field, see Table 1.

**Table 1.** Hall table for controlling reverse magnetic field.

| Status | Sector | Hall C | Hall B | Hall A | $S_1$ | $S_3$ | $S_5$ | $S_2$ | $S_4$ | $S_6$ |
|---|---|---|---|---|---|---|---|---|---|---|
| 1 | 5(VI) | 1 | 0 | 0 | 0 | PWM | 0 | 0 | 0 | 1 |
| 2 | 4(V) | 1 | 1 | 0 | PWM | 0 | 0 | 0 | 0 | 1 |
| 3 | 3(IV) | 0 | 1 | 0 | PWM | 0 | 0 | 0 | 1 | 0 |
| 4 | 0(I) | 1 | 0 | 1 | 0 | PWM | 0 | 1 | 0 | 0 |
| 5 | 1(II) | 0 | 0 | 1 | 0 | 0 | PWM | 1 | 0 | 0 |
| 6 | 2(III) | 0 | 1 | 1 | 0 | 0 | PWM | 0 | 1 | 0 |

We only focus on the braking operation in this research. Consider the first step of braking operation, the relay is cut off by the braking command, $S_8$ is disabled, and $S_7$ is enabled and the back EMF generated by the motor induces braking current is illustrated, as in Figure 4. The back EMF proceeds to charge the pump capacitor $C$ via the way indicated in the figure. Related variables and parameters are listed in Table 2.

**Table 2.** Nomenclature.

| Notation | Description |
|---|---|
| $V_m$ | voltage of the back EMF |
| $V_{ba}$ | battery voltage |
| $V_c$ | voltage of the pump capacitor |
| $V_{si}$ | drain-to-source voltage of the MOSFET $S_i$ |
| $i$ | current of circuit loop |
| $R$ | braking resistor |
| $R_{si}$ | body resistance of the MOSFET $S_i$ |
| $C$ | pump capacitor |

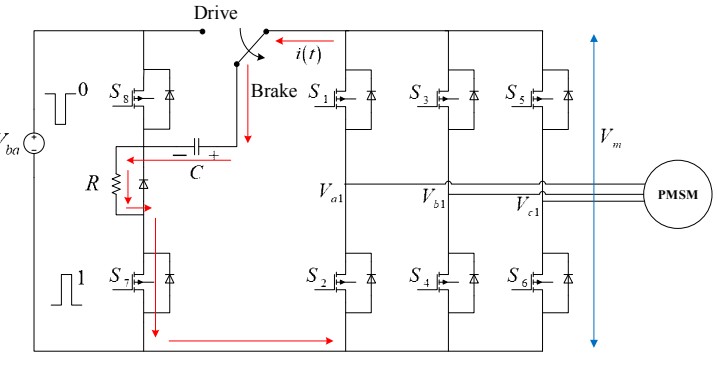

**Figure 4.** Capacitor charging loop.

From Figure 4, it is easy to have

$$V_m(t) - V_c(t) - i(t)R - V_{s7}(t) = 0 \tag{1}$$

where $V_c(t) = \frac{1}{c} \int i \, dt$ and $V_{s7}(t) = i(t)R_{s7}$ with $R_{s7}$ being the drain-to-source on-state resistance (body resistance) of the MOSFET $S_7$ and $V_m(t) = V_{emf} \sin(\omega t)$ where $\omega$ is the angular speed of the motor. The total impedance within the current loop is given by

$$\overline{Z} = R_{s7} + R - j\frac{1}{\omega C} = \sqrt{(R_{s7} + R)^2 + \left(\frac{1}{\omega C}\right)^2} \angle -\tan^{-1}\left(\frac{1}{\omega C(R_{s7} + R)}\right) \tag{2}$$

Let $\overline{V}_m = V_{emf} \angle 0°$ from

$$\overline{I} = \frac{\omega C V_{emf}}{\sqrt{[\omega C(R_{s7} + R)]^2 + 1}} \angle \tan^{-1}\left(\frac{1}{\omega C(R_{s7} + R)}\right)$$

From which we have the current

$$i(t) = \frac{\omega C V_{emf}}{\sqrt{[\omega C(R_{s7} + R)]^2 + 1}} \sin\left(\omega t + \tan^{-1}\frac{1}{\omega C(R_{s7} + R)}\right) \tag{3}$$

We use PWM commands to switch on/off status of $S_7$ and $S_8$ to charge the pump capacitor from the back EMF $V_m$. When $S_7$ is turned off and $S_8$ is turned on, the pump capacitor is cascaded with the battery, shown as in Figure 5.

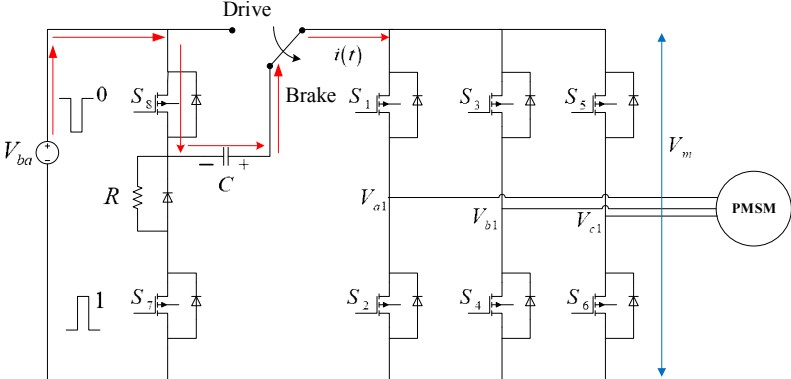

**Figure 5.** Current loop when the battery in series with the capacitor.

Under the situation, the battery voltage $V_{ba}$ will be in the series with the capacitor's voltage $V_c$. This boosts the main voltage to be $V_{ba} + V_c$. Referring to Figure 5, the braking current is the combination of the current generated by the DC voltage source $V_c(t)$, as well as the AC voltage source $V_m(t) = V_{emf} \sin(\omega t)$ with $V_{emf}$ being the amplitude of the back EMF.

We have the voltage loop equations corresponding to Status 2 in Table 1 given by

$$V_{ba} - V_{s8}(t) + V_c(t) - V_{s6}(t) - V_{s1}(t) - V_m(t) = 0 \tag{4}$$

where $V_{s8} = iR_{s8}$, $V_{s6} = iR_{s6}$, and $V_{s1} = iR_{s1}$. Analysis of the loop current $i(t)$ should consider the combination of two voltage sources.

Consider first the AC voltage source $V_m(t)$ generated by the back EMF. Clearly, the total impedance within the current loop is given by

$$
\begin{aligned}
\overline{Z}(j\omega) &= R_{s8} + R_{s1} + R_{s6} - j\frac{1}{\omega C} \\
&= \sqrt{(R_{s8} + R_{s1} + R_{s6})^2 + \left(\frac{1}{\omega C}\right)^2} \angle \tan^{-1}\left(\frac{-1}{(R_{s8}+R_{s1}+R_{s6})\omega C}\right)
\end{aligned}
$$

Let $\overline{V}_m = V_{emf}\angle 0°$. One can have

$$
\overline{I} = \frac{\overline{V}_m}{\overline{Z}(j\omega)} = \frac{\omega C V_{emf}}{\sqrt{[\omega C(R_{s8} + R_{s1} + R_{s6})]^2 + 1}} \angle \left(-\tan^{-1}\left(\frac{-1}{(R_{s8} + R_{s1} + R_{s6})\omega C}\right)\right)
$$

Therefore, the braking current contributed by the back EMF is given by

$$
i(t) = \frac{\omega C V_{emf}}{\sqrt{[\omega C(R_{s8} + R_{s1} + R_{s6})]^2 + 1}} \sin\left(\omega t - \tan^{-1}\left(\frac{-1}{(R_{s8} + R_{s1} + R_{s6})\omega C}\right)\right) \tag{5}
$$

Next, for DC voltage source, we have the node equation given by

$$
V_{ba} = V_{Rs8}(t) + V_{Rs1}(t) + V_{Rs6}(t) - (V_c(t) + V_c(0)) \tag{6}
$$

Or equivalently,

$$
V_{ba} + V_c(0) = i(t)(R_{s8} + R_{s1} + R_{s6}) + \frac{1}{C}\int i(t)dt
$$

This gives

$$
i(t) = \frac{V_{ba} + V_c(0)}{R_{s8} + R_{s1} + R_{s6}} e^{-\frac{1}{(R_{s8}+R_{s1}+R_{s6})C}t} \tag{7}
$$

Adding Equations (5) and (7) yields the total braking current given by

$$
i(t) = \frac{V_{ba} + V_c(0)}{R_{s8} + R_{s1} + R_{s6}} e^{-\frac{1}{(R_{s8}+R_{s1}+R_{s6})C}t} - \frac{\omega C V_{emf}}{\sqrt{[\omega C(R_{s8} + R_{s1} + R_{s6})]^2 + 1}} \sin\left(\omega t - \tan^{-1}\left(\frac{-1}{(R_{s8} + R_{s1} + R_{s6})\omega C}\right)\right) \tag{8}
$$

As $\omega C R_{si}$ is ignorable, from Equation (7) and the second term in Equation (8), it is seen that the braking current is proportional to the rotational speed of the motor, which is closely related to the value of $\omega$. Therefore, the electromagnetic braking effect would be decreasing with vehicle speed. The effect vanishes very fast when the vehicle is going to stop. This explains why the traditional design; simply with a braking resistance, does not work satisfactorily when the vehicle speed drops with time. Now consider the first term of Equation (8), because on-state resistance of the MOSFESTs $R_{si}$ are constant, it is seen that the braking effect contributed by both of the battery and back EMF would be longer when a large pump capacitor is used and the instantaneous braking current would be dominated by $V_{ba} + V_c(0)$. However, a large capacitor may not be fully charged to the supply voltage level if the duty cycle of the PWM command is not relatively large. Therefore, there is a trade-off for selection of an appropriate pump capacitor.

With the current $i(t)$, obtained from Equation (9), the charging equation of the pump capacitor can be obtained by the following equation where the sampling time period is $T$, the duty cycle is $D$ and the operating time is $t$:

$$
V_c(D, T, t) = \frac{1}{C}\int_{nT}^{(n+D)T} i(t)dt \tag{9}
$$

The discharging equation of the capacitor is given by

$$V_c(D,T,t) = \frac{1}{C} \int\limits_{(n+D)T}^{(n+1)T} i(t)dt \tag{10}$$

Therefore, the equation describing the pump capacitor's voltage over a sampling cycle of the PWM command is given by

$$V_c(D,T,t) = \frac{1}{C} \int\limits_{nT}^{(n+D)T} i(t)dt - \frac{1}{C} \int\limits_{(n+D)T}^{(n+1)T} i(t)dt + V_c(D,T,t-nT) \tag{11}$$

When $S_8$ is switched on, $S_1 \sim S_6$ are controlled, respectively, according to the reverse operating sequence of MOSFETs. By superposition, the back EMF used to generate braking current will be increased by the boosted voltage.

$$V_m(t) = V_{ba} + V_c(t) \tag{12}$$

This equation describes the total voltage used to reversely drive the motor within a sampling cycle and hence against wheel rotation for the largest stopping power. As the effect for wheel deceleration is created by adding extra energy rather than dissipating the back-EMF voltage via passive component, such as those of kinetic brake or dynamic brake, one could thus expect better braking performance to result.

Figure 6 illustrates the schematic diagram of the whole system where the controller/driver is implemented by a motor control 16-bit digital signal controller dsPIC30F3011 (MCU). Rotor position is detected by Hall sensors. The three-phase inverter is incorporated with the active regenerative braking control part, as shown in Figure 2. Brake signal is detected by a limit switch attached with the brake handle lever.

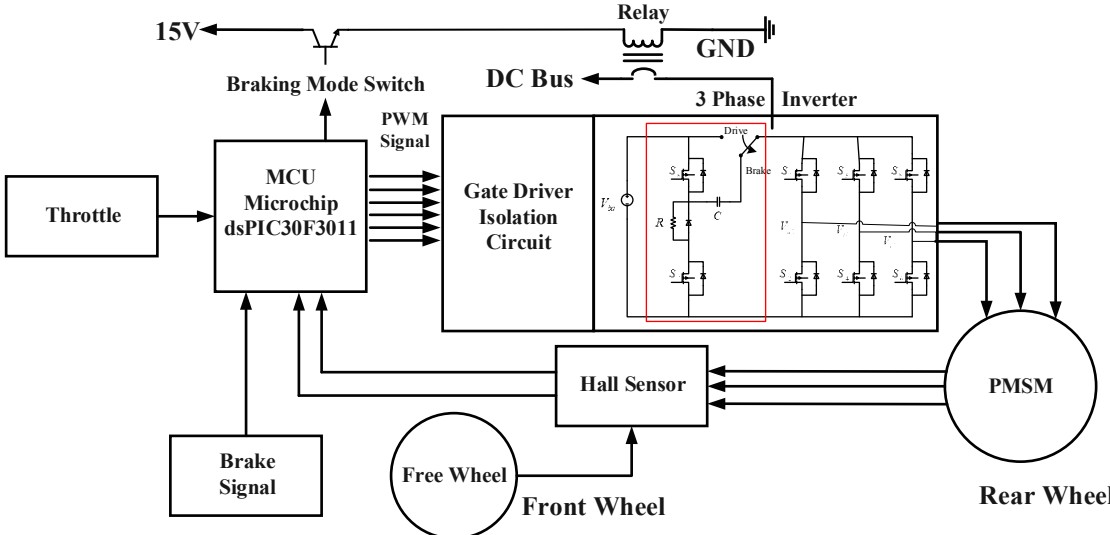

**Figure 6.** Schematic diagram of the whole system.

The MCU detects throttle and brake commands to determine which operating mode will be and count Hall signals to calculate the wheel speed. For driving, the MCU calculates the PID control command by comparing the throttle command with the wheel speed signal. This is used to alter the duty cycle of the pulse width modulated driving command of the six MOSFETs $S_1 \sim S_6$. While braking, the MCU sends command to cut off the current driving loop. The battery connects the pump capacitor, which is charged by the back EMF. The stacked voltage is then used to brake the motor in an active way

with the fixed duty cycle of the PWM driving command for the switching sequence of six MOSFETs, as depicted in Table 1.

## 3. Experimental Results Magnet

For the simulation purposes, considering the duty cycle $D$ being 75%, the sampling period $T$ is 0.2 msec, the resistance is 20 $\Omega$, the motor internal resistance $R_m$ is 5.6 $\Omega$, the motor inductance $L_m$ is is 22 mH, $R_{si} = 0$ for all $i$. We simulate charge and discharge statuses of the capacitor during braking. As mentioned, the MOSFETs $S_7$ and $S_8$ are controlled by the PWM command of 5000 Hz to charge and discharge the capacitor. Consider the back EMF to be 50 V. The PWM command uses 75% duty cycle for capacitor charging, and 25% duty cycle for discharging. Figures 7 and 8 illustrate the simulation results of the capacitor's voltage, where the capacitance is 20 $\mu$F and 200 $\mu$F, respectively, and the resistance of $R$ is 20 $\Omega$. Note that the peak voltage of the case of capacitance of 200 $\mu$F is lower than that of 20 $\mu$F because the current setting of duty cycle of the PWM command is not enough for the larger capacitor fully charged to the supply voltage.

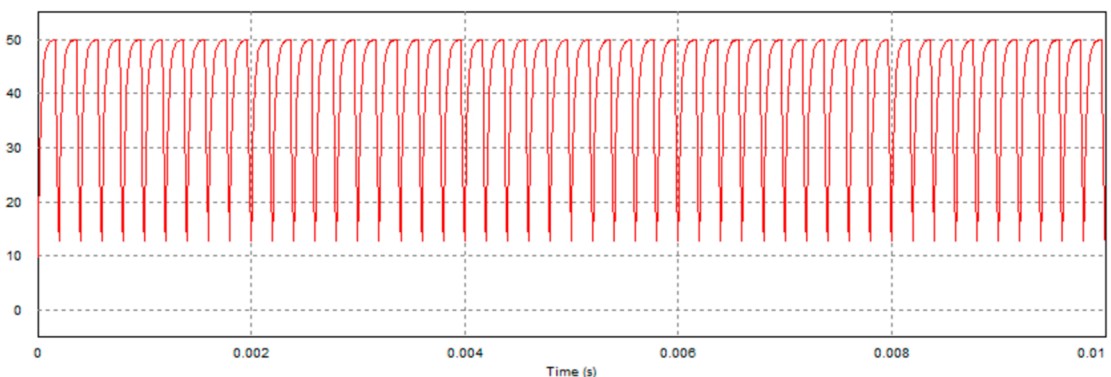

**Figure 7.** Simulating charge and discharge voltage of 20 $\mu$F capacitor.

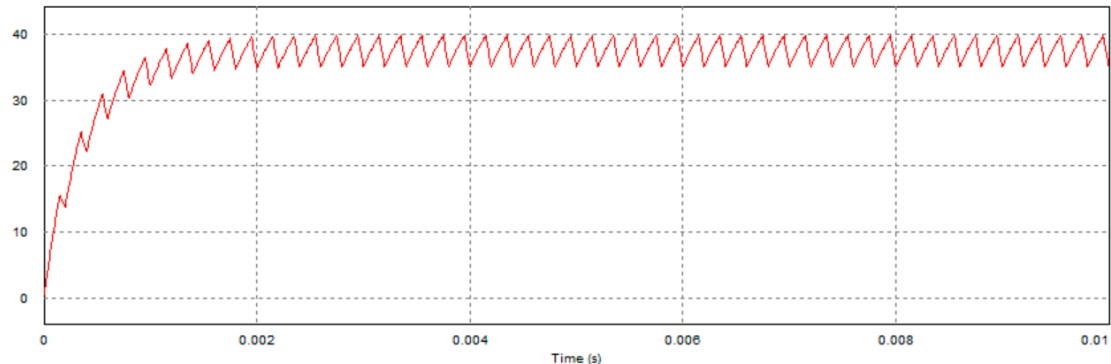

**Figure 8.** Simulating charge and discharge voltage of 200 $\mu$F capacitor.

*Experimental Verification*

The electric bicycle was tested on the TACX I-Genius T2000 bike simulator [17]. The simulator is designed as a virtual reality experimental platform for bike trainers. The interactive trainer is equipped the Tacx Trainer software where users can choose from various animated terrains for mountain bike, race, and track with virtual 3D riders. Several data are provided like bike speed, time, road slope, distance, and power. What we need in this research work are just speed, time, slope, and distance. We perform hardware-in-the-loop (HIL) experiments by including all hardware, software, and the simulator with respect to two kinds of road condition including smooth flat road and smooth downhill road of a −5% slope. The arrangement helps us to understand the behavior of the proposed braking control design in an efficient way. In which, the bike simulator is equipped with an active resistance

unit driven by a DC motor to generate the effect of riding uphill or downhill. During a hill climbing, the resistance at the rear wheel increases accordingly. While going downhill, the speed of the active resistance unit will be increased to emulate the riding downhill scenarios. The testing environment can be found from Figure 9. We verify braking performance of the active regenerative braking control system by implementing it in a customized electric bicycle. The bicycle has a drum direct-drive motor with 20 poles and three 120° Hall sensors installed in the rear wheel. The friction brake has been removed from the rear wheel. Driving and braking processes are controlled solely by electricity.

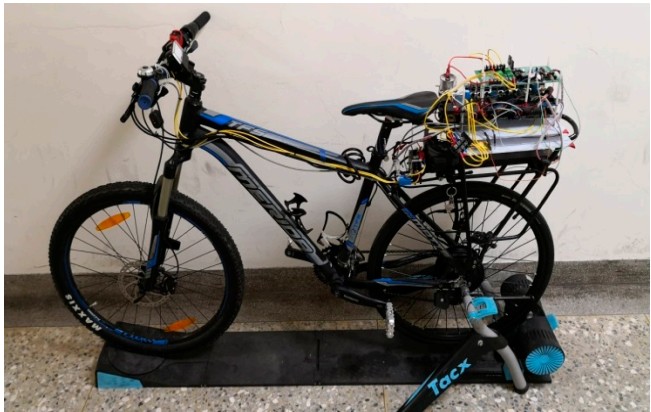

**Figure 9.** Experimental platform of electric bicycle.

An active regenerative braking control system has been equipped with a pump capacitor of 200 μF, as depicted in Section 2. Charge or discharge of the capacitor depends on the way of activation. Figure 10 illustrates the capacitor's voltage when charging and discharging. RMS value of the capacitor's voltage is around 75 V. The battery is of 48 V. The extra voltage is stacked to the battery voltage to boost the braking current in an active way so as to enhance braking torque.

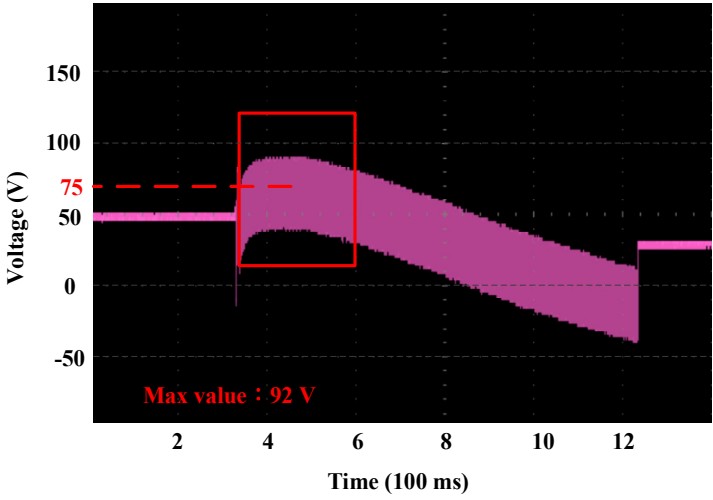

**Figure 10.** Response of capacitor voltage.

For high-speed inertial rotation of the motor, it generates larger back EMF. While adding the back EMF to the fully charged capacitor, the instantaneous voltage is boosted to the peak value of 123 V, without the need of external power, as shown in Figure 11.

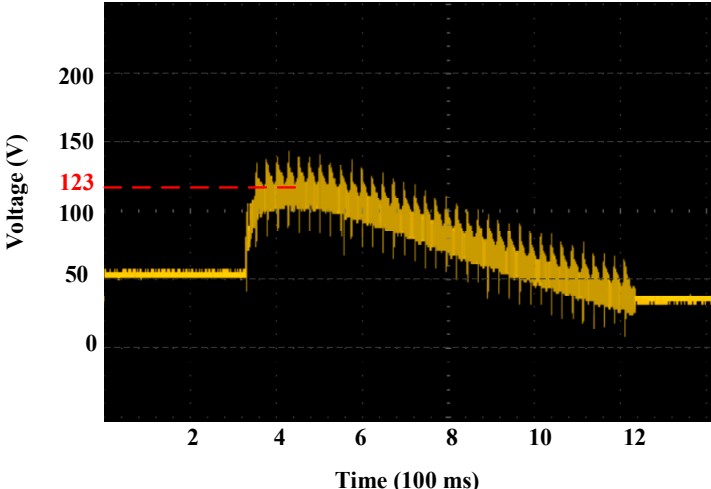

**Figure 11.** Boosted voltage when the battery is in series with the pump capacitor.

Figure 12 shows braking current of the active regenerative braking control system, where the peak current reaches 40 A, which significantly increases while compared with the non-active regenerative braking methods.

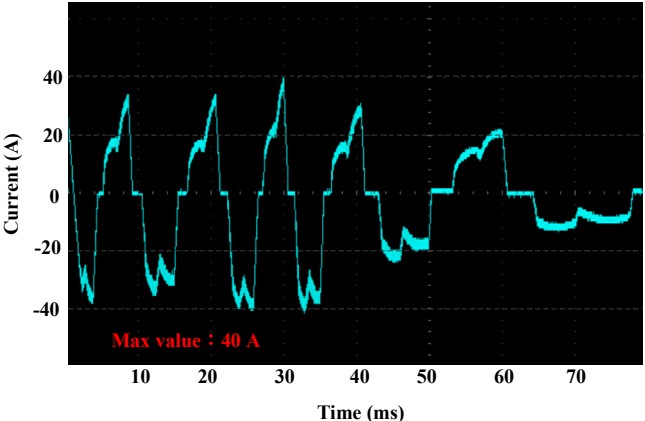

**Figure 12.** Braking current of the active regenerative braking control system.

We used the simulator to measure the bicycle speed. The simulation results were verified by conducting the test on the smooth flat road surface. The total traveling distance in all experiments was set to be 100 m. The bicycle was speed up to 25 km/h and last for 20 s. Then, the braking command was executed without the aid of the mechanical brake. The stopping distance was used to identify the level of effectiveness of the braking system. We endeavored to maintain the sufficiently consistent wheel rotational speed for fair comparisons. Electromagnetic braking was activated after the bicycle speed reaches 25 km/h.

A 60 kg payload to emulate the rider with the same weight to conduct the test on the smooth flat road. We verified braking performance (in stopping distance) of different operational modes, namely, kinetic brake with the virtual load of 1 Ω, dynamic brake with the virtual load of 20 μF, and an active regenerative brake. Once the bicycle speed approaches 25 km/h and lasts for 20 s, the selected braking mode was applied.

For the kinetic braking mode, the stopping distance was 21.8 m shown as in Figure 13.

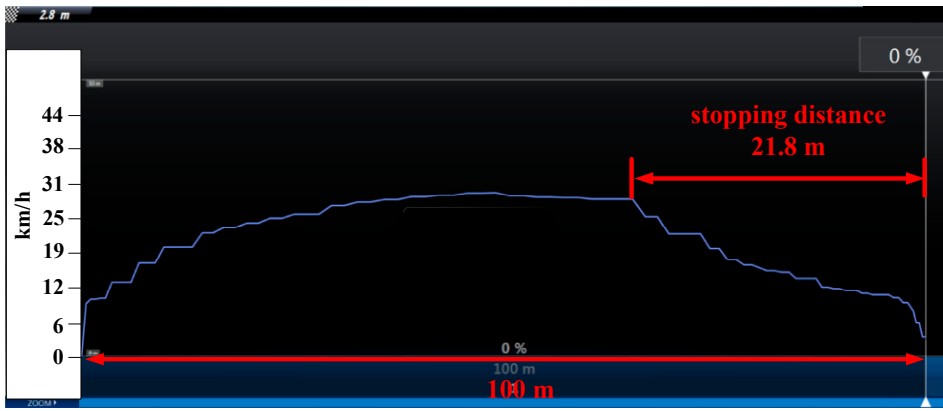

**Figure 13.** HIL test result of rider of 60 kg, flat road with kinetic brake.

For the dynamic braking mode, the stopping distance was measured as 13.3 m (in Figure 14).

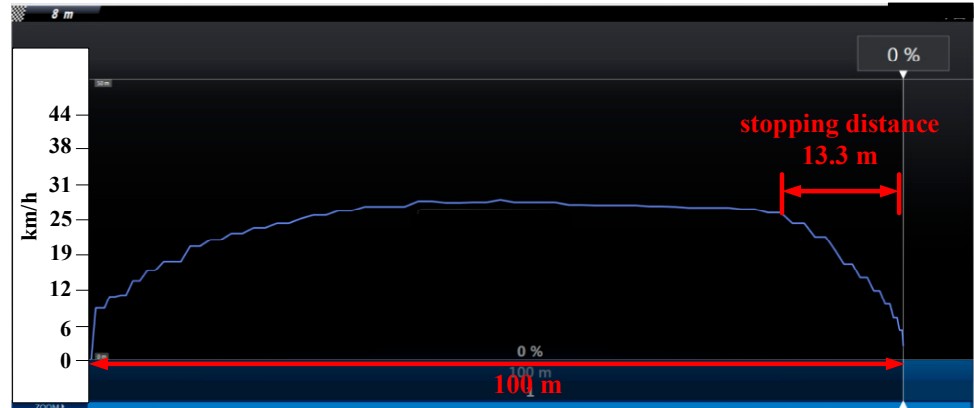

**Figure 14.** HIL test result of rider of 60 kg, flat road with dynamic brake.

The stopping distance of the active regenerative braking system was 10.3 m (in Figure 15). It is seen that the stopping distance of the active regenerative braking system is shorter than that of the dynamic brake by 3 m. Obviously, the active regenerative braking control system shows superior performance than other designs.

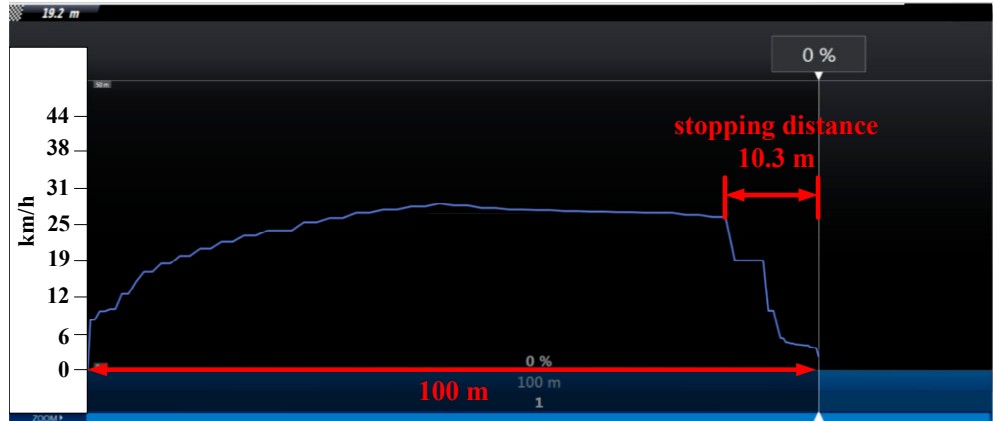

**Figure 15.** HIL test result of rider of 60 kg, flat road with active regenerative brake.

To verify effectiveness of the proposed design, various scenarios were considered. In each of the scenario, specific changes of the rider weight and road condition were set as a contingency to the

system. Two kinds of road condition including smooth flat road and smooth downhill road of a −5% slope are considered. The results of HIL test, represented in terms of the key index, e.g. the stopping distance, are summarized in Table 3. Under the current settings, the braking performance of the active regenerative brake surpasses the other two approaches by 30% to 50%.

**Table 3.** Summary of HIL test results.

| Initial Speed (km/h) | Weight (kg) | Road Conditions | Brake Method | Stopping Distance (m) |
|---|---|---|---|---|
| 25 | 60 | smooth flat | Kinetic brake | 21.8 |
| | 60 | smooth flat | Dynamic brake | 13.3 |
| | 60 | smooth flat | Active regenerative brake | 10.3 |
| | 80 | smooth flat | Kinetic brake | 26.8 |
| | 80 | smooth flat | Dynamic brake | 19.2 |
| | 80 | smooth flat | Active regenerative brake | 17.5 |
| 30 | 60 | smooth downhill of −5% slope | Kinetic brake | 23.7 |
| | 60 | smooth downhill of −5% slope | Dynamic brake | 15.9 |
| | 60 | smooth downhill of −5% slope | Active regenerative brake | 12.3 |
| | 80 | smooth downhill of −5% slope | Kinetic brake | 28.3 |
| | 80 | smooth downhill of −5% slope | Dynamic brake | 21.4 |
| | 80 | smooth downhill of −5% slope | Active regenerative brake | 19.5 |

Video demonstration of the real-world experiments on the smooth road surface has been uploaded to YouTube. Please visit the website in Reference [18] for braking performance demonstration.

## 4. Discussions

The experimental results listed here for a 60 kg rider revealed that, compared with other braking system designs, the proposed active regenerative braking control system presents superior performance in terms of the stopping distance with its values reduced up to 22%. For the smooth flat road with a rider of 80 kg in weight, the proposed design exhibits superior performance with the stopping reduced by 10%. As for the smooth downhill road with the slope 5% and the rider of 60 kg weight, the proposed braking control system reduces the stopping distance up to 23%. The same test scenario for the rider of 80 kg show that the stopping distance was reduced by 9%.

Theoretical analysis and experimental results reveal that the stopping distance was considerably reduced by the application of a virtual load with lower resistance or even zero $\Omega$. However, this may induce the risk of overcurrent leading to excessive generation of heat and bring up the risk of damage to MOSFETs. Therefore, there is a compromise in selecting an appropriate virtual load in the braking current loop.

In addition, our design is mainly to provide an assistant to the mechanical brake, especially when the bike is riding downhill (rather than to completely replace it). As the proposed design can easily tune the strength of braking force by adjusting duty cycle of the pulse-width modulated command, it can be used to generate the effect of engine brake like those of the gas-powered vehicles for safety concern.

## 5. Conclusions

This research task proposes a novel active regenerative braking control design. The design has been implemented, in both hardware and firmware, for real-world experiments. In the presented design, the back EMF is charged into a pump capacitor to boost the energy for braking torque enhancement. This is used to generate larger braking current and pursue stronger braking torque. The proposed system has been successfully implemented in the electric bicycle to demonstrate its effectives, while compared with the previous approaches, which were simply based on the use of the self-induced, back-EMF voltage from the motor. Durability testing has also been conducted on the bike simulator to test the limit of the system capability against large braking current under various load conditions over time.

As for the future works, a sensor-less electromagnetic braking control system based on the field-oriented control (FOC) to integrate the driving and the braking units into a unified controller is currently under development, which allows one to properly size motors and drives, lowering cost and resulting in a more efficient system overall. In addition, the development of anti-lock braking system (ABS); based on the current braking design scheme; is worthy of further study.

**Author Contributions:** Conceptualization, C.-L.L.; Methodology, C.-L.L. and H.-C.H.; Software, H.-C.H.; Validation, H.-C.H.; Formal Analysis, H.-C.H. and J.-C.L.; Investigation, C.-L.L. and H.-C.H.; Data curation, H.-C.H.; Resources, C.-L.L.; Writing—Original Draft Preparation, H.-C.H.; Writing—Review & Editing, C.-L.L.; Visualization, H.-C.H.; Project Administration, C.-L.L.; Funding Acquisition, C.-L.L.

**Funding:** This research was sponsored by Ministry of Science and Technology, Taiwan under the grant MOST 104-2221-E-005-093-MY2. This work was also supported in part by the Innovation and Development Center of Sustainable Agriculture (IDCSA) under the Higher Education Sprout Project, MOE in Taiwan.

**Conflicts of Interest:** The authors declare no conflict interest.

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
