# Peer review of "Active Control of Regenerative Brake for Electric Vehicles"

_actuators, doi:10.3390/act7040084_

Reviewer 1 Report
The article is about an Active Control of Regenerative Brake for Electric Vehicles. I think that the article is interensting in electric vehicles. I have the following comments:
The general structure of the article must be improved, the introduction section is short. There are many regenerative control system that you could include in it. The discussion and conclusion sections could also improve.
The Figures should be changed o improved.Figures 9,10,11,12,13 and 14 are not clear, have chinese caracters and it is imposible to see labels and so on.
I do not understands the dynamic brake and kinetic brake modes, I think you should explain better. There is a new braking mode called Resistance Braking System in Table 2. What does it mean?
I think you should show the stopping distance when you use the friction brake. Is it shorter?
Author Response
We deeply appreciate the reviewers for their valuable comments and suggestions, which greatly helped us to improve the paper’s quality. The following descriptions are our responses to the reviewers’ comments. Necessary changes have been highlighted in green color in the revised manuscript.
Reviewer: 1
The article is about an Active Control of Regenerative Brake for Electric Vehicles. I think that the article is interesting in electric vehicles. I have the following comments:
1. The general structure of the article must be improved, the introduction section is short. There are many regenerative control system that you could include in it. The discussion and conclusion sections could also improve.
Ans.: Thank you for the comment. More references have been added in the Introduction. Discussion and Conclusion have also been enhanced.
2. The Figures should be changed improved. Figures 9,10,11,12,13 and 14 are not clear, have Chinese caracters and it is impossible to see labels and so on.
Ans.: Thank you for the comment. Figures 9,10,11,12,13 and 14 have been redrawn for better readability.
3. I do not understand the dynamic brake and kinetic brake modes, I think you should explain better. There is a new braking mode called Resistance Braking System in Table 2. What does it mean?
Ans.: Thank you for the comment. Dynamic brake and kinetic brake have been explained in the Abstract. We have changed the term of Resistance Brake to be kinetic bake in Table 2.
4. I think you should show the stopping distance when you use the friction brake. Is it shorter?
Ans.: Thank you for the suggestion. We didn’t include friction brake for comparison. Because braking effect based on friction brake would be dependent on how large the rider’s strength used to pull the brake handle lever, thus it’s uneasy to set up a fair standard for comparison. In general, friction brake would be shorter in stopping distance if the brake handle lever is fully pulled. However, the proposed design is better in agility and suitable for being use as that of engine brake in the gas powered vehicles.
Reviewer 2 Report
Although the proposed study is promising, there are some concerns regarding the details of the presented method as well as the organization of paper. Furthermore, the structure of the submitted manuscript and the literature survey requires some improvements before publication of the paper.
The reviewer is curious to see the attempts of the authors in revising the paper and answering the questions about the proposed algorithm. The reviewer has the following suggestions/comments:
1)The abstract can be revised to include more details of the proposed approach. Authors may improve the abstract by including the existing challenges, motivations and outcomes of the paper.  Abstracts usually have at least one sentence per each: context and background, motivation, hypothesis, methods, results, conclusions.
2)As there are a large number of variables, parameters, and sets, authors can potentially provide nomenclatures in the first section of paper. It helps the reader to follow the paper conveniently.
3)Introduction section is incomplete. It mainly can include four key components: motivation, literature survey, contributions, and the organization of paper. Please modify this section accordingly.
4)Figures require improvement in terms of the quality and font type. It is recommended to the respected authors to use the same font size for all figures.
5)The simulation results can be extended to support the proposed approach effectively. Authors need to add more illustrative figures as well as some comparison tables to clearly highlight the key findings by simulation results.
6)In order to make the proposed method understandable, authors are encouraged to add a schematic overview of the whole paper.
7)Please talk about the future work briefly in the conclusion section.
Author Response
We deeply appreciate the reviewers for their valuable comments and suggestions, which greatly helped us to improve the paper’s quality. The following descriptions are our responses to the reviewers’ comments. Necessary changes have been highlighted in green color in the revised manuscript.
Reviewer: 2
1. The abstract can be revised to include more details of the proposed approach. Authors may improve the abstract by including the existing challenges, motivations and outcomes of the paper.  Abstracts usually have at least one sentence per each: context and background, motivation, hypothesis, methods, results, conclusions.
Ans.: Thank you for the suggestion. The Abstract has been rewritten under your direction.
2. As there are a large number of variables, parameters, and sets, authors can potentially provide nomenclatures in the first section of paper. It helps the reader to follow the paper conveniently.
Ans.: Thank you for the comment. We have included Table 2 to summarize critical variables and parameters used in this paper.
3. Introduction section is incomplete. It mainly can include four key components: motivation, literature survey, contributions, and the organization of paper. Please modify this section accordingly.
Ans.: Thank you for the comment. Introduction has been partly rewritten under your direction.
4. Figures require improvement in terms of the quality and font type. It is recommended to the respected authors to use the same font size for all figures.
Ans.: Thank you for the comment. We have checked all figures and made necessary amendments.
5. The simulation results can be extended to support the proposed approach effectively. Authors need to add more illustrative figures as well as some comparison tables to clearly highlight the key findings by simulation results.
Ans.: Thank you for the comment. We have provided comparison results in Table 3 that has covered almost all key findings of simulation results with respect to different rider weight, road condition, brake method and initial speed. Real-world comparison results with respect to different brake methods can also be watched from [17] via the video clip on Youtube. Because of the strict limit of page length, only the representative HIL test results of 60 kg rider was provided in Figs. 12-14. However, we are happy to attach more figures if the reviewer insists.
6. In order to make the proposed method understandable, authors are encouraged to add a schematic overview of the whole paper.
Ans.: Thank you for the suggestion. We have added a schematic diagram (Fig. 6) to describe the overall system.
7. Please talk about the future work briefly in the conclusion section.
Ans.: Thank you for the suggestion. Future works have been added in the conclusion.
Reviewer 3 Report
Contribution of the work is rather clear. Writing style must be improved nevertheless.
For example, some equations (e.g. in paragraph after (1) ) are not in line with text, they should be properly aligned to text.
"During a hill climb or go uphill" and "hill descent or going downhill": what is the difference between hill climbing and going uphill? same with descent and downhill.
- different font and size is used after Table 1, than with the rest of the manuscript.
Assuming braking tests were performed with some non-slipping surface, authors do mention that test conditions were "under various designed road conditions", later being described as "Two kinds of road condition including smooth flat road and smooth downhill road of -5% slope are considered." But which are the surfaces conditions? please state clearly. This is mainly since braking depends substantially on contact surface.
- how does the proposed ABS control system of this work relate to similar existing work? Literature review should be enhanced with some recent titles: Braking sense consistency strategy of electro-hydraulic composite braking system, A Pressure-Coordinated Control for Vehicle Electro-Hydraulic Braking Systems, Regenerative Braking Control Strategy of Electric-Hydraulic Hybrid (EHH) Vehicle, Data-driven model-free slip control of anti-lock braking systems using reinforcement Q-learning, Nonlinear adaptive controller applied to an Antilock Braking System with parameters variations.
- authors must proofread the paper to a native english speaker, to improve readability.
- especially with respect to regenerative braking which is a highly researched topic, authors must better emphasize the novelty of their work.
- to make the results more clear, details on the computing hardware being used with TACX I_genius T2000 should be provided.
Revision of this work is further advised.
Author Response
We deeply appreciate the reviewers for their valuable comments and suggestions, which greatly helped us to improve the paper’s quality. The following descriptions are our responses to the reviewers’ comments. Necessary changes have been highlighted in green color in the revised manuscript.
Reviewer: 3
1. Contribution of the work is rather clear. Writing style must be improved nevertheless. For example, some equations (e.g. in paragraph after (1) ) are not in line with text, they should be properly aligned to text.
Ans.: Thank you for the comment. We have checked the whole manuscript and made necessary corrections.
2. "During a hill climb or go uphill" and "hill descent or going downhill": what is the difference between hill climbing and going uphill? same with descent and downhill.
Ans.: Thank you for the comment. Hill climbing is identical to going uphill. Hill descent is identical to going downhill. These terms have been unified in the revised paper.
3. Different font and size is used after Table 1, than with the rest of the manuscript.
Ans.: Thank you for the comment. The problem has been fixed.
4. Assuming braking tests were performed with some non-slipping surface, authors do mention that test conditions were "under various designed road conditions", later being described as "Two kinds of road condition including smooth flat road and smooth downhill road of -5% slope are considered." But which are the surfaces conditions? please state clearly. This is mainly since braking depends substantially on contact surface.
Ans.: Thank you for the comment. Inappropriate statement has been rewritten in this revised paper. See Section 3.
5. How does the proposed ABS control system of this work relate to similar existing work? Literature review should be enhanced with some recent titles: Braking sense consistency strategy of electro-hydraulic composite braking system, A Pressure-Coordinated Control for Vehicle Electro-Hydraulic Braking Systems, Regenerative Braking Control Strategy of Electric-Hydraulic Hybrid (EHH) Vehicle, Data-driven model-free slip control of anti-lock braking systems using reinforcement Q-learning, Nonlinear adaptive controller applied to an Antilock Braking System with parameters variations.
Ans.: Thank you for the helpful suggestion, five recent papers have been referred in the revised paper. We didn’t realize ABS control in this work (which is not the theme in this research task), however, our design can be extended to incorporate the function.
8. C. Wang, W. Zhao and W. Li, “Braking sense consistency strategy of electro–hydraulic composite braking system,” Mechanical Systems and Signal Processing, 2018, pp. 196-219.
9. Y. Yang, G. Li and Q. Zhang, “A Pressure-Coordinated Control for Vehicle Electro-Hydraulic Braking Systems,” Energies, 2018.
10.Y. Yang, C. Luo and P. Li, “Regenerative Braking Control Strategy of Electric-Hydraulic Hybrid (EHH) Vehicle,” Energies, 2017.
11.M. B. Radac and R. E. Precup, “Data-driven model-free slip control of anti-lock braking systems using reinforcement Q-learning,” Neurocomputing, 2018, vol. 275, pp. 317-329.
12.C. A. Lua, S. D. Gennaro and M. E. S. Morales, “Nonlinear adaptive controller applied to an Antilock Braking System with parameters variations,” International Journal of Control, Automation and Systems, 2017, vol. 15, pp. 2043-2052.
6. Authors must proofread the paper to a native English speaker, to improve readability.
Ans.: Thank you for the comment. The paper has been revised again for typos and grammatical check.
7. Especially with respect to regenerative braking which is a highly researched topic, authors must better emphasize the novelty of their work.
Ans.: Thank you for the comment. Novelty of our approach has been highlighted and explained in the Introduction.
8. To make the results more clear, details on the computing hardware being used with TACX I_genius T2000 should be provided.
Ans.: The interactive trainer has provided the Tacx Trainer software [16]. Users can choose from various animated terrains for mountain bike, race and track with virtual 3D riders. There are several data like bike speed, time, road slope, traveling distance and power dissipation provided by the trainer. We just collect related data from the software without doing further development. What we need here is only bike speed, running time, road slope and traveling distance. This has been explained in the revised manuscript.
Reviewer 4 Report
The paper is very interesting and well-written. It is hard to find any pros when the quality of presentation is concerned. Nevertheless, I have a few comments related to general topics:
1) abstract - last sentence - it would be good to describe the criterion used to discriminate among solutions, more in detail
2) Figure 3 - BRAKING SINGLE is not explained well enough in the body of the paper
3) notation of (1) and below is not explained in the text - maybe it is possible to use a table to present all abbreviations and notation at once?
4) please mind the typos with subindices, as below (1) Vm -> V_M and so on
5) above (2) RESISTANCE should be changed rather to IMPEDANCE
6) sampling idea is not introduces well enough in the text (from (9) and below), and it is vital in the context of an overall behaviour of the system
7) Authors use km/h or km/hr notation - please standarize this
8) section 4: ohm has been used in the text with the symbolic \Ohm.
Author Response
We deeply appreciate the reviewers for their valuable comments and suggestions, which greatly helped us to improve the paper’s quality. The following descriptions are our responses to the reviewers’ comments. Necessary changes have been highlighted in green color in the revised manuscript.
Reviewer: 4
1. Abstract - last sentence - it would be good to describe the criterion used to discriminate among solutions, more in detail.
Ans.: Thank you for the comment. The last sentence in the Abstract has been rewritten. See the revised Abstract.
2. Figure 3 - BRAKING SINGLE is not explained well enough in the body of the paper.
Ans.: Thank you for the comment. The term is incorrect and has been removed.
3. Notation of (1) and below is not explained in the text - maybe it is possible to use a table to present all abbreviations and notation at once?
Ans.: Thank you for the comment. Table 2 summarizing critical variables and parameters have been added in the revised paper.
4. Please mind the typos with subindices, as below (1) Vm -> V_M and so on.
Ans.: Thank you for the comment. The paper has been carefully checked to correct all typos.
5.  Above (2) RESISTANCE should be changed rather to IMPEDANCE.
Ans.: Thank you for the suggestion. It has been corrected.
6.  Sampling idea is not introduces well enough in the text (from (9) and below), and it is vital in the context of an overall behavior of the system.
Ans.: Thank you for the comment. The explanation has been provided at the paragraph under Eq. (11).
7.  Authors use km/h or km/hr notation - please standardize this.
Ans.: Thank you for the correction. It has been fixed.
8.  Section 4: ohm has been used in the text with the symbolic \Ohm.
Ans.: Thank you for the correction. It has been fixed.
Round  2
Reviewer 1 Report
I think that the article have improved and it can be accepted for publication.
Author Response
We deeply appreciate again the reviewers for their valuable comments and suggestions, which greatly helped us to improve the paper’s quality. The following descriptions are our responses to the reviewers’ comments. Necessary changes have been highlighted in blue color in the re-revised manuscript.
Review comments:
1. In my opinion it is enough if you comment more in depth scheme nummer 6. In partcular considering the interaction of all elements present there. It is also to consider to move this diagram before the discussion and explanation of the singel parts. The idea should be to give an overview of the complete scheme before to discuss each singular part. This could help the reader.
Ans.: Thank you for the comment. More explanations have been added under your direction.
2. With respect to the literature it could be useful to consider the following paper and the literature therein which can help to have a wider overview on the current literature on this Topic. In fact, a problem of the control an electric City-Bus is consider in which is also visible a braking phase.
Ans.: Thank you for the comment. The reference has been added in the Introduction.
Reviewer 2 Report
The revised version reads well.
Author Response

(The authors gave the same response as above.)
